# Can a Greenhouse Gas Emissions Tax on Food also Be Healthy and Equitable? A Systemised Review and Modelling Study from Aotearoa New Zealand

**DOI:** 10.3390/ijerph19084421

**Published:** 2022-04-07

**Authors:** Christine Cleghorn, Ingrid Mulder, Alex Macmillan, Anja Mizdrak, Jonathan Drew, Nhung Nghiem, Tony Blakely, Cliona Ni Mhurchu

**Affiliations:** 1Burden of Disease Epidemiology, Equity and Cost-Effectiveness Programme, Department of Public Health, University of Otago, Wellington 6021, New Zealand; anja.mizdrak@otago.ac.nz (A.M.); nhung.nghiem@otago.ac.nz (N.N.); 2Department of Preventive and Social Medicine, University of Otago, Dunedin 9016, New Zealand; mulin062@student.otago.ac.nz (I.M.); alex.macmillan@otago.ac.nz (A.M.); jonodrew1@gmail.com (J.D.); 3Population Interventions, Centre for Epidemiology and Biostatistics, Melbourne School of Population and Global Health, University of Melbourne, Melbourne, VIC 3010, Australia; antony.blakely@unimelb.edu.au; 4National Institute for Health Innovation, University of Auckland, Auckland 1142, New Zealand; c.nimhurchu@auckland.ac.nz; 5The George Institute for Global Health, Newtown, NSW 2042, Australia

**Keywords:** GHG emissions, food taxes, nutritional epidemiology, review, simulation modelling

## Abstract

Policies to mitigate climate change are essential. The objective of this paper was to estimate the impact of greenhouse gas (GHG) food taxes and assess whether such a tax could also have health benefits in Aotearoa NZ. We undertook a systemised review on GHG food taxes to inform four tax scenarios, including one combined with a subsidy. These scenarios were modelled to estimate lifetime impacts on quality-adjusted health years (QALY), health inequities by ethnicity, GHG emissions, health system costs and food costs to the individual. Twenty-eight modelling studies on food tax policies were identified. Taxes resulted in decreased consumption of the targeted foods (e.g., −15.4% in beef/ruminant consumption, N = 12 studies) and an average decrease of 8.3% in GHG emissions (N = 19 studies). The “GHG weighted tax on all foods” scenario had the largest health gains and costs savings (455,800 QALYs and NZD 8.8 billion), followed by the tax—fruit and vegetable subsidy scenario (410,400 QALYs and NZD 6.4 billion). All scenarios were associated with reduced GHG emissions and higher age standardised per capita QALYs for Māori. Applying taxes that target foods with high GHG emissions has the potential to be effective for reducing GHG emissions and to result in co-benefits for population health.

## 1. Introduction

Climate change is a major threat to human civilisation and health [1] and is increasingly recognised as a determinant of wellbeing and increasing inequities [2]. The urgent need to limit global warming by reducing greenhouse gas (GHG) emissions has gained widespread international acceptability following the United Nations Framework Convention on Climate Change and adoption of the Paris Agreement. There is a strong case for interventions targeting agricultural production as food systems are responsible for up to 29% of global anthropogenic GHG emissions [3]. One mechanism for this is through policies to change what people consume. Three quarters of global agricultural GHG emissions are associated with meat production, through land use change and enteric methane emissions [4]. Although improvements to livestock farming methods to reduce the associated GHG emissions are likely to be implemented in the future [5], there will still need to be global reductions in meat production and consumption alongside these innovations to reduce emissions.

The consumption of red and processed meat is associated with an increased risk of chronic disease [6,7,8,9]. Fruits, vegetables, nuts and seeds are associated with a decreased risk of coronary heart disease (CHD), stroke, type 2 diabetes and various cancers [10]. Increased consumption of fruits, vegetables and wholegrains and reduced red and processed meat consumption could therefore reduce the detrimental effects to our long-term health and reduce the burden on the health care system [11]. Research has also demonstrated the potential climate co-benefits of these dietary changes [12].

Taxes have been shown to reduce consumption of harmful products, such as alcohol [13], tobacco [14] and sugar-sweetened beverages [15], and evidence shows that the more products a tax covers the more effective it is in changing people’s purchasing behaviours. It is also important to consider any potentially regressive effects of a tax and to design the tax to reduce negative outcomes, in terms of both the financial impact on households and health inequities. While GHG food taxes are internationally regarded as potential instruments to help achieve emissions reduction [16], clear gaps in the evidence-base exist [17,18]. The appropriateness and efficacy of taxes are likely to vary across different contexts, and the different ways food taxes can be designed and implemented will also affect the impact of a tax.

This paper aimed to first review the international literature on the effects of food taxes motivated by reducing GHG emission. We then used this evidence to design a variety of taxes aimed at reducing the purchasing, and therefore consumption, of high-GHG-emitting foods. These global taxes were tailored to the New Zealand context, which is high-income with diets high in animal products and above average per capita GHG emissions. It forms a good case study for other countries with diets high in animal products. These taxes were modelled to estimate their impact on population health gain (in quality-adjusted life-years: QALYs), health system cost-savings, health inequities between Māori (NZ indigenous population) and non-Māori, GHG emissions and food costs. This wide range of outcomes will allow policymakers to consider the impacts of these taxes on health, health inequities and the climate and to make more informed policy decisions. At present NZ-specific evidence for the effect of GHG food taxes on these outcomes is lacking [19,20,21].

## 2. Materials and Methods

### 2.1. Literature Review

We took a systemised approach to reviewing studies on tax policies motivated by reducing GHG emissions; the query string used within each database combined search terms relating to four themes: climate change; food systems; tax policies; and health. We searched Scopus and Web of Science (WoS), limited to the past 10 years (2010–2020, search run on 8 December 2020) and those studies written in English. Four independent searches (in Scopus and Web of Science with and without the “health theme”) were carried out. (See the Appendix A for additional details including Appendix A for the search strategy.) An iterative process was used to refine the search strategy to produce the final search terms (Appendix A).

Both modelling studies and real-world evaluations were included if they:(a)Illustrated a tax motivated by reducing GHG emissions.(b)Allocated a quantifiable tax amount to a defined food group or groups.

Studies were excluded if they:(a)Included only a tax on foods which was not motivated by reducing GHG emissions.(b)Did not allocate a quantifiable tax amount to a defined food group or groups.(c)Only assessed environmental outcomes other than GHG emissions, e.g., biodiversity loss.(d)Only related to subsidy policies.(e)Only included a tax that was not specific to food groups (e.g., fuel tax or electricity tax).

All references of included studies after initial title/abstract screening were screened for inclusion. Studies within reviews identified were also screened against the exclusion and inclusion criteria. We extracted the following data from included studies: location, study design, tax justification/purpose, tax description and quantity, targeted food group/type, and outcomes including changes in price, amount of food produced and/or consumed, GHG emissions, impacts on health disparities, health system costs and population health.

### 2.2. GHG Tax Scenarios

Next, the results of the literature review were used to design a selection of taxes to be modelled in the New Zealand context. Among the various GHG food tax approaches proposed within the literature, four were selected for further investigation and corresponding scenarios were designed on the basis that they provided a range of options to help understand the impact of differently designed taxes.

The taxes designed for modelling were designed primarily to reduce GHG emissions, while exploring their potential for achieving health co-benefits and reducing health inequities between Māori and non-Māori. To explore the potentially regressive impacts of food taxes, we designed one that attempts to minimise the overall financial impact of taxes on consumers by providing a compensatory subsidy for fruit and vegetables. The scenarios were informed by the results of the literature review and details of selected taxes are presented in the Section 3.

### 2.3. Modelling Methods

**Dietary intake data**: Dietary intake data were sourced from the most recent representative New Zealand Adult National Nutrition Survey (NZANS) [22] and used as an estimate for baseline intake. This was conducted in 2008/09 (data acquired from the University of Otago’s Life in New Zealand Research Group who conducted the survey, through personal communication, Blakey, Smith and Parnell, 2014). Dietary data were from a single 24 h dietary recall and are in grams (g) per food group for each of 338 food groups. Average intakes per food group were calculated for sex by ethnic groups (Māori and non-Māori).

We used an existing NZ-specific price elasticity matrix [23] disaggregated into a 338 by 338 food group matrix to align with consumption data. We constrained total food expenditure by using total food expenditure elasticity (TFEe) of 0.75 [24,25]. These methods are well-established and are consistent with earlier work modelling the impact of food taxes in NZ. Further detail is presented in the Appendix A and elsewhere [26,27,28,29].

**Modelling:** In this modelling, GHG taxes change the price of foods which changes consumption of specific foods. This change in consumption impacts GHG emissions and, as it changes the intake of dietary risk factors, it also goes on to impact disease incidence and therefore QALYs. This modelling is therefore a useful tool to assess health and climate co-benefits of food taxes.

The tax scenarios were applied to the baseline diet (the business-as-usual (BAU) comparator) in the dietary intervention model and impact the unit price of food items. The taxes influenced food purchases through price elasticities, which subsequently affected consumption. Differences in food consumption between BAU and the tax scenarios were simulated for the entire New Zealand population, alive in 2011 (N = 4.4 million), using an Excel-based dietary proportional multi-state life-table model (PMSLT) populated with high quality epidemiological data [30,31] and using established methods [32,33,34,35]. Outputs from this modelling were changes in daily GHG emissions per person in kgCO_2_-eq, percentage change in daily cost of diet per person, the price index of the diet (a measure of relative price changes of the total diet) and the following outputs over the life course of those alive in 2011: incremental population QALYs gained; ethnic health inequities (ratio of age adjusted per capita health gains between Māori and non-Māori) and costs or cost-savings to the health system in New Zealand dollars (NZD). Detailed modelling methods are included in the Appendix A. See Appendix A for baseline input parameters.

**GHG emissions:** The units for the carbon taxes are tonnes of kgCO_2_-eq whereby CO_2_-eq refers to a comparable unit that averages GHG emissions to equal the same global warming impact as CO_2_, using the standard GWP100 method (GWP100 is the accounting metric adopted by the Intergovernmental Panel on Climate Change in inventory guidelines); this allows for comparison between products with different levels of GHGs.

A New-Zealand-specific life-cycle assessment (LCA) database was previously developed by modifying cradle to point-of-sale reference emissions estimates from an established UK database to the New Zealand context. This UK database, presented in Hoolohan et al. (2013) [36] provided per-kg cradle to point-of-sale emissions estimates for 66 food categories. It included the relative contributions of the following life-cycle stages: farming and processing; transportation; transit packaging; consumer packaging; warehouse and distribution; refrigeration; and supermarket overheads. Each NZANS food group used in modelling was matched to an NZ-specific LCA (if available) or a food category from the reference LCA database and emissions estimates were assigned accordingly.

UK emissions estimates were modified to the NZ context, with efforts focused on life-cycle stages that contributed most to overall emissions and those where the NZ context was expected to differ most from the UK database (transportation and electricity usage). Further details on these methods are outlined in Drew et al., 2020 [37].

CO_2_-eq per 100 g of food group was used to calculate the amount of tax applied to each food group. To calculate the threshold to be used to define high-GHG-emitting foods (henceforth referred to as the “high emitters”), we averaged the CO_2_-eq per 100 g of food for the 338 food groups in the NZANS: 0.46 kgCO_2_-eq/100 g.

**Sensitivity and further scenario analyses:** Māori are disadvantaged in the main analysis as they have higher background morbidity and mortality; this results in a lesser “envelope” for potential health gains. In order to value health gain in Māori the same as for non-Māori, an equity analysis was modelled in which background morbidity and mortality rates for Māori were set to non-Māori values [38]. All scenarios were rerun with no discounting so health gain in the future is valued the same as health gain in the present. A lower and upper sensitivity analysis was carried out for all scenarios modelled.

## 3. Results

### 3.1. Review Results

A total of 3564 records were identified across the four searches conducted (see Figure 1). Following review, there were 28 included studies, of which 27 were modelling studies, with 1 cost-benefit modelling analysis [39]. The locations of these studies were the UK (5) [17,40,41,42,43], international (4) [44,45,46,47], Spain (4) [48,49,50,51], the EU (3) [52,53,54], France (3) [55,56,57], Sweden (2) [58,59], the Netherlands [39], Switzerland [60], Denmark [18], Canada [61], Australia [62], Norway [63] and Belgium [4]. Seven studies also included a subsidy on food groups [18,28,34,37,40,46,50]. Seven had a fixed percentage tax on food products [39,42,43,49,51,56,63]. The remaining 21 studies had calculated tax rates based on a carbon price per unit of food [2,4,17,46,60,64]. The carbon taxes ranged 25,000-fold from 0.01 GBP/tCO_2_-eq (0.02 NZD) [42] to 290 EUR/tCO_2_-eq (489.92 NZD) [52]. The food groups targeted ranged from all food groups to just two groups [45,47,51]. There were 14 studies which only taxed animal products [39,42,43,44,45,47,49,52,53,55,56,57,59,61], two of which also included a scenario that taxed all products [42,57]. Appendix A summarises the key characteristics and results of the included studies.

Prices increased mainly for animal products, except in Dogbe et al. [48] where fish/seafood prices decreased by 5% and 15% when tax revenues generated from the taxed foods were used to subsidise lower-emission foods under a 56 EUR/tCO_2_-eq (94.54 NZD) and 200 EUR/tCO_2_-eq (337.65 NZD) tax, respectively. In the 21 publications reporting percentage change in the price of beef, the average price change was 28.1% (range −8.3% [51] to 90.1% [45]). The average increase in price for studies that targeted dairy products was 22.5% (N = 14, range 1.9% [11] to 113.7% [52]).

Generally, modelling studies showed that taxing foods based on their GHG emissions decreases beef/ruminant consumption, with an average decrease of 15.4% (N = 12 studies reporting % change in consumption) with changes ranging from an increase in consumption of 10.4% [51] to a decrease of 49.0% [63]. Due to cross-price elasticities (that represent changes in purchases of food groups that are not taxed, e.g., if they are substitutes for the taxed foods), some modelling studies showed an increase in consumption of non-alcoholic drinks and fresh fruit; pork and poultry; and snacks and other foods [48,53,62]. Of nine studies that examined dietary energy intake, eight saw a decrease [17,18,30,35,39,44,46,51], the largest being a 14.9% reduction [57].

GHG emissions generally decreased when a tax was modelled. One study designed their tax to meet specific GHG emission reduction targets [63]. All other studies showed modelled reductions in GHG emissions ranging from 0.4% [54] under a 50 USD/tCO_2_-eq (69.46 NZD) tax to 19.4% [18] under a 760 DKK/tCO_2_-eq (174.80 NZD) tax. The average of the 19 papers reporting percentage change in GHG emissions for the relevant jurisdiction was −8.3% (when multiple scenarios were reported, the largest change was included in this calculation). In contrast, Broeks et al. [39], who modelled a 10% subsidy on fruit and vegetables in the Netherlands, found an increase of 4.5% in environmental impact based on GHG emissions, acidification, water eutrophication and land use.

Vandenberghe et al. [4] observed a saving of up to 79,800 disability-adjusted life years (DALY) for a tax of 60 EUR/tCO_2_-eq (101.88 NZD) in the modelled year in Belgium. A higher tax on all foods (96 CHF per tCO_2_-eq (151.02 NZD)) estimated a reduction of 706 DALYs per year for the Swiss population [60]. Springmann et al. [62] estimated a reduction of 49,500 DALYs or 1620 averted deaths at a tax rate of 23 AUD/tCO_2_-eq (24.66 NZD) in Australia. In total, 7770 deaths were modelled to be averted in a UK modelling study (tax of 2.72 GBP/tCO_2_-eq (5.17 NZD)) but when this tax was modelled in combination with a subsidy for low-GHG-emission foods, an extra 2685 deaths occurred [17]. However, lower estimates of deaths delayed or averted were seen with a similar modelled tax (2.86/tCO_2_-eq (5.44 NZD)) alone (300), combined with a subsidy on low-emission foods (90), combined with a 20% SSB tax (1200) or combined with both the subsidy and SSB tax (2000) [40]. Springmann et al. [46] estimated 107,000 avoided deaths at a global scale, with a tax rate of 52 USD/tCO_2_-eq (72.24 NZD).

Only two of the included studies commented on the health system impacts of a modelled GHG emission food tax. Vandenberghe et al. [4] suggested that a 30, 45 and 60 EUR/tCO_2_-eq (50.72, 76.11 and 101.88 NZD) tax in Belgium could save EUR 256–EUR 481 million in the modelled year in terms of medical expenditure. Broeks et al. [39] reported that meat taxes of 15% or 30%, or a 10% fruit and vegetable subsidy could save up to EUR 7.4, 12.3 and 3.3 billion over 30 years, respectively.

### 3.2. Modelling Results

#### 3.2.1. Modelled GHG Food Tax Scenarios

The following four tax scenarios to be modelled in the New Zealand context were designed based on the results of the literature review.

**S1: GHG weighted tax, all foods:** Firstly, we have chosen a conceptually simple approach where all foods are taxed based on their GHG emissions. This approach has been taken in previous modelling studies [41,48,60] with the levels of taxation varying between 5.40 NZD per tCO_2_-eq [41] and 337.65 NZD per tCO_2_-eq [48].

The approach taken for this paper was to set the tax level so the ANS dietary food group “beef, muscle meat” (chosen as it is 100% meat rather than a composite food group such as casserole) increased in price by 20%. This is a relatively arbitrary tax level often chosen in tax modelling and advocacy; as such, we carried out sensitivity analyses which corresponded to 10% and 40% for each scenario. The amount of tax necessary to increase the price of “beef, muscle meat” by 20% was 163.59 NZD/tCO_2_-eq/100 g of food. This amount was applied to all food groups in the ANS and is applied per 100 g of food to line up with the structure of the model. Values used in the S1 (lower) and S1 (upper) scenarios were 82.03 NZD/tCO_2_-eq/100 g of food and 327.18 NZD/tCO_2_-eq/100 g of food.

**S2: GHG weighted tax, “high emitters”:** The second scenario is the same as the first but targets “high emitters” (emissions above 0.46 kgCO_2_-eq/100 g). As in S1, the magnitude of the tax is weighted by the GHG emissions of each food group (163.59 NZD/tCO_2_-eq/100 g). This modelling approach, targeting high-emitting foods only, was taken in Bonnet et al. [55] and in one of the scenarios presented in Dogbe et al. [48]. Values used in the S2 (lower) and S2 (upper) scenarios were the same as in S1.

**S3: GHG weighted tax and subsidy:** Thirdly, we combined the weighted tax outlined for S2 on “high emitters”, with a 20% subsidy on all fruit and vegetables. S3 aims to be approximately price neutral to reduce the negative impact on household finances of regressive taxes. Previous modelling studies have used subsidies in combination with taxes to partially offset their financial impacts on individuals [17,18,57], with value-added tax being removed on all foods [18] or subsidies being applied to low-emitting foods [17] and fruits, vegetables and starchy foods [57]. Values used in the S3 (lower) and S3 (upper) scenarios were the same as in S1 for the tax and 10% for S3 (lower) and 40% for S3 (upper) for the subsidy.

**S4: Percentage tax on “high emitters”**: As the first three scenarios may be administratively difficult to implement, we modelled a more straightforward proxy. We applied a set percentage tax on the “high emitters”. This approach is similar to several other modelling studies, which taxed meat (beef, pork and chicken [61], beef and sheep meat [45,47] or animal products [42,49,52]). Taxes have been set to a range of percentages or set amounts per tCO_2_-eq for specific high-GHG-emitting foods [45,47,52,61] and one modelling study presented both approaches [42]. A 20% tax was chosen following the same rationale as S1–S3. This was the same as in Caillevat et al. [56] and was one of the taxes used in Revoredo-Giha et al. [42]. Values used in the S4 (lower) and S4 (upper) scenarios were 10% and 40% taxes. See Appendix A for tax rates and target food groups for all scenarios.

#### 3.2.2. Impacts on Modelling Outcomes

**S1:** Average changes in the dietary risk factors which have an impact on disease incidence were as follows: BMI decreased by 0.5 kg/m^2^; red and processed meat intake decreased by 10 g/p/d with small decreases in vegetable and sodium intake. Small increases were seen in intakes of fruit, sugar-sweetened beverages, polyunsaturated fat and nuts (see Table 1). The largest changes in food group intake, compared to baseline, were a decrease of “milk” (−22 g) and “grains and pasta—rice only” (−20 g) (Appendix A).

These changes in intake led to the largest health gains (432,000 QALYs gained over the remaining lifespan of the NZ population alive in 2011, or 98 QALYs gained per 1000 people alive in 2011) and cost savings (NZD 8.2 billion) of all scenarios (Table 2). Age standardised per capita health gain for Māori was 1.8 times that of non-Māori. This ratio increased to 2.3 when the equity analysis was modelled. Health gain was higher for men than for women (ratio of 1.3). S1 also generated the greatest GHG emission reductions compared to the baseline NZ diet, approximately 0.35 kgCO_2_-eq per person per day or a 7.0% reduction in average diet-generated GHG emissions (Table 3). Cost of diet increased by 3.8% with a change in the price index of 5.0%.

**S2:** There was a small decrease in BMI (−0.1 kg/m^2^), red meat and processed meat (−11 g/p/d) and sodium (−25 mg/p/d) and small increases in fruit and vegetables (8 g/p/d) and SSBs (3 g/p/d). The largest decreases in food group intake, compared to baseline, were a decrease in “beef & veal” (−5 g), “sausage & processed meats” (−4 g) and “bread-based dishes” (−4 g).

Health gains and cost savings were approximately half that of S1 (196,900 QALYs and NZD 3.6 billion). Age standardised per capita health gain for Māori was 2.1 times, and 2.7 times when the equity analysis was applied, that of non-Māori. Health gain was higher for men than for women (ratio of 1.4). S2 saved approximately 0.21 kgCO_2_-eq of GHG emissions per person per day (4.3% of the baseline diet). Cost of diet increased by 2.5% with a change in the price index of 1.9%.

**S3:** BMI decreased by an average of 0.2 kg/m^2^, red and processed meat by 15 g/p/d and sodium by 51 mg/p/d. Fruit and vegetable intake increased by 28 g and 50 g, respectively. The largest changes in food group intake, compared to baseline, were a decrease of “non-alcoholic beverages” (−24 g) alongside the increases in fruit and vegetable intake.

An increase of 410,400 QALYs was seen with associated cost-savings to the health system of NZD 6.4 billion. Age standardised per capita health gain for Māori was 1.6 times that of non-Māori and 2.2 times higher when the equity analysis was modelled. Health gain was higher for men than for women (ratio of 1.2). The difference in consumption between the baseline diet and S3 saved an average of 0.21 kgCO_2_-eq of GHG emissions per person per day (4.2% of baseline diets). Cost of diet decreased by 0.5% with a change in the price index of −0.8%.

**S4:** There was an increase in fruit and vegetable intake (20 g) and SSB intake (7 g) and a small average increase in BMI of 0.03 kg/m^2^. Sodium decreased by 35 mg/p/d and red and processed meat by 13 g/p/d. The largest changes in food group intake, compared to baseline, were a decrease in “bread-based dishes” (−13 g) and “fish and seafood” (−5 g).

QALY gains were 152,800 and health system cost savings were NZD 2.4 billion. Age standardised per capita health gain for Māori was 1.7 times that of non-Māori and 2.3 times higher when the equity analysis was modelled. Health gain was 1.6 times higher for Māori women than Māori men but was 1.1 times higher for non-Māori men than for non-Māori women. S4 saved approximately 0.22 kgCO_2_-eq of GHG emissions per person per day (4.5% of baseline diets). Cost of diet increased by 4.7% with a change in the price index of 6.2%.

**Sensitivity and further scenario analyses:** Table 3 presents the sensitivity and further scenario analysis results. Health gain was approximately 3.7 times greater when results were not discounted over time for all scenarios. Health gain in the lower scenarios (equivalent to a 10% increase in the price of “beef, muscle meat”) was between 0.5 and 0.7 of the base case health gain for all scenarios. Health gain in the upper scenarios (equivalent to a 40% increase in the price of “beef, muscle meat”) was between 0.4 of the health gains (due to an increase in BMI in S4 (upper)) and 1.7 times greater than the base case scenarios. Health system cost savings were approximately 2.7 times greater when undiscounted but otherwise mirrored these patterns. GHG savings in the lower scenarios were approximately half of the base case scenarios. GHG savings in the upper scenarios were between 1.4 (S3) and 2.0 (S2) times greater than the base case scenarios.

Figure 2 plots health gain against GHG emission reductions to visually represent health and climate co-benefits and includes the main four scenarios with their upper and lower sensitivity analyses. S1 (upper) gives the most health gain and GHG emission reductions of all taxes by a clear margin. This is followed by S3 scenarios and then S2 scenarios. S4 is the only set of scenarios which does not show a linear trend between the lower, main and upper scenarios, with the health gain for S4 (upper) being lower than for S4 (lower). This is due to the change in BMI being −0.03, 0.03 and 0.37 for S4 (lower), S4 and S4 (upper), respectively.

Appendix A shows QALYs and costs/cost savings that occur in the first 10 and 20 years of the taxes. Between 4% and 5% of lifetime QALYs accrue in the first ten years and between 18% and 19% in the first twenty years of the taxes. Cost savings were between 9% and 12% of lifetime cost savings and between 30% and 34% in the first ten and twenty years of the taxes.

## 4. Discussion

**Main findings:** The review found 28 modelling studies of food tax policies motivated by reducing GHG emissions. Most of these studies calculated tax rates based on a carbon price per unit of food using a wide range of tax rates. Studies consistently showed that the increased price of targeted foods decreased their consumption and decreased total GHG emissions.

We developed a range of tax scenarios based on the review. The largest health gains and cost savings occurred in the scenario where a tax was applied to all foods, weighted by GHG emissions, closely followed by the tax subsidy scenario. The changes in dietary risk factors responsible for these health gains and cost savings were increases in fruits and vegetables (up to almost 80 g for the tax subsidy scenario) and small decreases in red and processed meat and sodium. Most of the health gain, however, was due to the decrease in average BMI across the population from the decrease in the energy content of the taxed diets. All scenarios were associated with GHG savings (between 4.2% and 7.0% of baseline diets).

Indigenous Māori in NZ are disproportionately affected by the health effects of diet, particularly foods and nutrients linked to cardiovascular disease and type 2 diabetes [65]. This is a result of both higher consumption and wider structural injustices. NZ has an obligation under te Tiriti o Waitangi (the constitutional Treaty between the British Crown and Māori) to support Māori health, and policies that affect diets need to consider how they can reduce health inequities between Māori and non-Māori. Health gain was between 1.6 and 2.1 times higher for Māori than non-Māori for these scenarios (using age standardised per capita QALYs) so these taxes have the potential to reduce ethnic health inequities.

It is also important to consider impacts on household costs when considering tax policy design. Food costs to the individual increased by between 1.9% and 4.7% for the tax-only scenarios and decreased by 0.5% for the tax subsidy scenario. The GHG tax on all food scenario has the largest health gain, cost savings, GHG impact and second highest potential impact on health equity. However, combining a GHG tax with a fruit and vegetable subsidy may be the best approach to employ, considering the large health gain and cost savings, reductions in GHG emissions and food costs, improvements in fruit and vegetable consumption and health equity and practical implementation considerations.

**Strengths and limitations:** The dietary data for the modelled baseline consumption are taken from the latest Adult National Nutrition survey, carried out in 2008/09, and consumption in New Zealand may have changed over the last 12 years. Data from the NZ health survey show a steady decrease in adult fruit and vegetable consumption over this time period (https://minhealthnz.shinyapps.io/nz-health-survey-2020-21-annual-data-explorer/ accessed on 1 February 2022). The modelled impacts on health may therefore be conservative. The base year of the PMSLT model used was 2011, with trends on disease incidence, case fatality and remission out until 2026. Disease rates may have changed since 2011, potentially altering the impact of these taxes.

There is uncertainty around the extent to which price elasticities can predict future changes in purchasing which, in this modelling, is applied to consumption data to estimate the change in BMI. The price elasticities matrix that we used was generated using Bayesian priors from previously published New Zealand food price elasticities [28]. The applied TFEe method aimed to limit implausible changes in food intake that may be generated through breaches in econometric assumptions of PE estimation.

This study uses CO_2_ equivalents as the metric for GHG emissions. There is an argument to be made that methane should be reported separately to carbon dioxide and nitrous oxide due to their different length of warming potentials, and the specific methane targets developed at the global level (e.g., “Global Methane Pledge” at the COP26 U.N. Climate Summit, 2021). Three of the modelled taxes have targeted food groups with high GHG emissions, those with more than the average CO_2_-eq per 100 g of food for the 338 food groups in the NZANS. This does not take into consideration the amount of these foods consumed in NZ and it may be more efficient to focus taxes on commonly consumed high emitters. This was not explored in this paper.

**Policy implications:** The results of the review clearly show positive impacts on consumption and GHG emissions of modelled GHG food taxes, though these would not be enough alone to meet obligations for food emissions reductions. The modelling shows health and climate co-benefits from four different GHG food pricing policies. There would be practical challenges in translating these taxes to a real-world setting, including likely opposition from food producers and industry, how GHGs would be measured, the metrics used to equivalise methane, and how product- and producer-specific the tax should be.

It also needs to be acknowledged that there are particular limitations in NZ to taxing domestic consumption where a large proportion of high-emitting foods are produced for export (e.g., 89% of bovine meat and 87% of mutton and goat meat in 2019 [66]). Much greater reductions in GHG emissions would likely be achieved by targeting agricultural production in NZ. Equity implications of taxing individual consumers rather than pricing emissions at the point of production are also important to consider. However, considering the scope and urgency of the reductions in GHG emissions needed, multiple policies that have synergistic effects on food production and consumption will need to be implemented, ideally including policies with health co-benefits such as a consumption tax on high-GHG-emitting foods. The money raised through the tax could be redirected to encourage green technologies and the money saved from reduced chronic disease could be used to make improvements within the health system. Additionally, food producers remain sensitive to their domestic consumers, and taxing food consumption may increase social pressure to address GHG emissions from total agricultural production.

## 5. Conclusions

The results of this article illustrate that applying consumption taxes that target foods with high GHG emissions associated with their production has the potential to result in population health and climate co-benefits. The four different tax designs all generated reductions in GHG emissions alongside health gains and could therefore be a policy governments could employ to meet both health and environmental targets. Importantly for the New Zealand context, these taxes also have the potential to reduce ethnic health inequities. Cost savings to the health system and tax revenue could be directed to expand these benefits. Whilst NZ is used as a case study, findings are likely to be of relevance to other high-income countries. Policymakers need to weigh up the benefits of such a consumer food tax against other options that are likely to result in improved diets and reduced food system GHG emissions. A consumption tax combined with a subsidy on fruit and vegetables could be considered to minimise the impact on household food costs.

## Figures and Tables

**Figure 1 ijerph-19-04421-f001:**
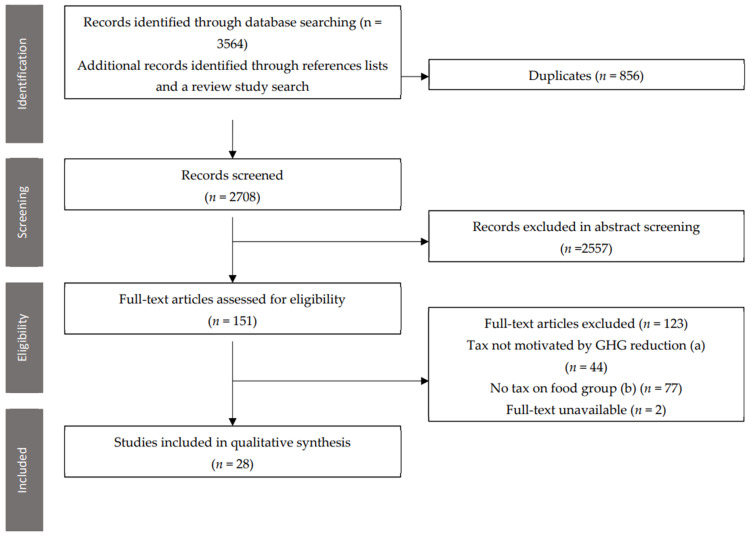
A PRISMA flow diagram following the process of study identification and eligibility screening.

**Figure 2 ijerph-19-04421-f002:**
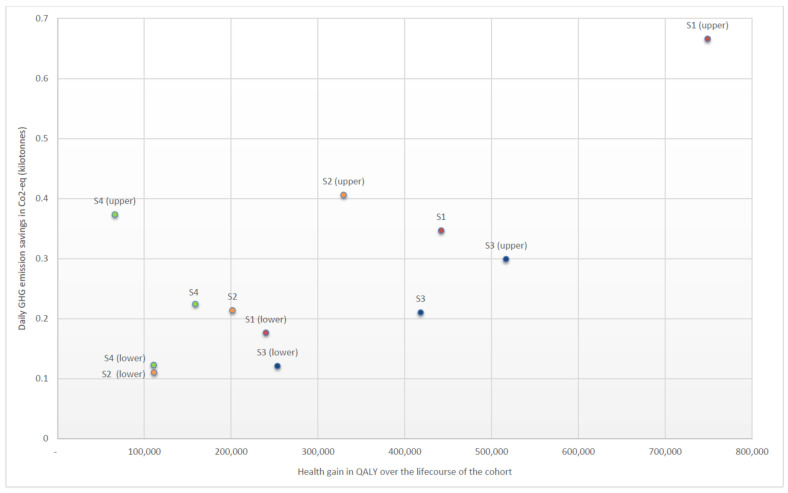
GHG emission impact against health impact of the four main scenarios and their upper and lower sensitivity analyses.

**Table 1 ijerph-19-04421-t001:** Population weighted change in BMI and dietary risk factors used for modelling of taxes.

	∆ BMI	∆ Fruit (g/day)	∆ Vegetables (g/day)	∆ Red Meat (g/day)	∆ Processed Meat (g/day)	∆ SSB (g/day)	∆ Nuts and Seeds (g/day)	∆ Sodium (mg/day)	∆ PUFA (% TE)
S1: GHG weighted tax, all foods	−0.48	3.2	−1.6	−4.1	−5.5	1.0	0.2	−52.2	0.1%
S2: GHG weighted tax, high emitters	−0.10	3.8	4.0	−5.5	−5.9	2.8	0.2	−25.4	0.0%
S3: GHG weighted tax and subsidy	−0.19	28.4	50.4	−7.4	−7.3	−0.1	−0.1	−50.8	0.0%
S4: Percentage tax on high emitters	0.03	9.7	10.2	−4.7	−7.8	7.0	0.5	−34.6	0.0%

BMI: body mass index; SSB: sugar-sweetened beverage; TE: total energy intake.

**Table 2 ijerph-19-04421-t002:** Lifetime health impacts (in QALYs) and health system costs for GHG food taxes, for the NZ population alive in 2011 (lifetime horizon) with 3% discount rate.

	Non-Māori	Māori	Māori	Ethnic Groups Combined	
	Health Gains: QALYs	Health Gains: QALYs	Equity Analysis [38] Health Gains: QALYs	Health Gains: QALYs	Net Health System Cost Savings (NZD Billion)
S1: GHG weighted tax, all foods					
Total	327,300 (226,500 to 467,800)	104,700 (70,700 to 153,400)	138,100 (94,100 to 202,200)	432,000 (298,200 to 615,000)	NZD 8.2 (5.4 to 12.4)
Men	184,000	58,200	77,100	242,200	NZD 4.7
Women	143,300	46,400	61,000	189,700	NZD 3.6
Per capita *	87.7 (113.6)	155.2 (201.3)	204.9 (266.4)	98.1	NZD 1866.8
S2: GHG weighted tax, highest emitters					
Total	143,700 (92,000 to 219,500)	53,200 (32,400 to 84,000)	69,700 (44,200 to 107,600)	196,900 (125,000 to 303,000)	NZD 3.6 (2.1 to 5.8)
Men	84,700	28,800	37,900	113,500	NZD 2.1
Women	59,000	24,400	31,800	83,300	NZD 1.5
Per capita *	38.5 (49.6)	78.9 (102.4)	103.4 (134.4)	44.7	NZD 816.6
S3: GHG weighted tax and subsidy					
Total	322,000 (259,800 to 391,300)	88,400 (74,000 to 104,900)	118,900 (99,400 to 141,400)	410,400 (336,200 to 492,000)	NZD 6.4 (4.9 to 8.2)
Men	175,500	44,500	59,800	220,000	NZD 3.6
Women	146,500	44,000	59,100	190,400	NZD 2.8
Per capita *	86.3 (106.5)	131.2 (170.6)	176.3 (229.6)	93.2	NZD 1451.8
S4: Percentage tax on highest emitters					
Total	118,500 (4900 to 296,200)	34,300 (−8300 to 100,100)	46,600 (−8000 to 126,900)	152,800 (−3000 to 396,600)	NZD 2.4 (−0.7 to 7.2)
Men	63,200	13,200	18,500	76,400	NZD 1.2
Women	55,200	21,100	28,200	76,400	NZD 1.2
Per capita *	31.7 (39.9)	50.9 (66.7)	69.2 (90.7)	34.7	NZD 549.8

* Per capita results: QALYs/1000 people and NZD/person.

**Table 3 ijerph-19-04421-t003:** Health, health system cost and GHG emission impacts of sensitivity and further scenario analyses around the base case for each scenario.

Sensitivity/Scenario Analyses	Health Gains: QALYs (Millions)	Net Health System Cost Savings (NZD Billion)	Change in kgCO_2_-eq per Person per Day	Cost of Diet per Person per Day (% Change from Baseline Diets)
S1: GHG weighted tax, all foods				
Base case analysis *	0.44	NZD 8.5	−0.35	3.8%
S1 (lower)	0.24	NZD 4.6	−0.18	1.9%
S1 (upper)	0.75	NZD 14.2	−0.67	7.6%
Undiscounted	1.62	NZD 23.7		
S2: GHG weighted tax, high emitters				
Base case analysis *	0.20	NZD 3.7	−0.21	1.9%
S2 (lower)	0.11	NZD 2.1	−0.11	1.0%
S2 (upper)	0.33	NZD 5.9	−0.41	3.9%
Undiscounted	0.74	NZD 10.0		
S3: GHG weighted tax and subsidy				
Base case analysis *	0.42	NZD 6.5	−0.21	−0.5%
S3 (lower)	0.25	NZD 4.1	−0.12	−0.3%
S3 (upper)	0.52	NZD 6.7	−0.30	−1.0%
Undiscounted	1.55	NZD 16.8		
S4: Percentage tax on high emitters				
Base case analysis *	0.16	NZD 2.5	−0.22	4.7%
S4 (lower)	0.11	NZD 1.9	−0.12	2.4%
S4 (upper)	0.07	NZD 0.0	−0.37	9.5%
Undiscounted	0.59	NZD 6.5		

* 3% discounting, no discounting applied to GHG emissions.

## Data Availability

Access to the dietary data used in this modelling study was provided by Statistics New Zealand under conditions designed to keep individual information secure in accordance with requirements of the Statistics Act 1975. The opinions presented are those of the author(s) and do not necessarily represent an official view of Statistics New Zealand. Access to the GHG emissions data used in this modelling is available in the Appendix A of a previously published article [37].

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
