# Peer review of "Can a Greenhouse Gas Emissions Tax on Food also Be Healthy and Equitable? A Systemised Review and Modelling Study from Aotearoa New Zealand"

_ijerph, 2022, doi:10.3390/ijerph19084421_

Round 1
Reviewer 1 Report
The paper is very interesting and can be published after some minor adjustments. Suggestions:
- In the introduction: I suggest to the authors show some evidences about the relationship between less consumption of red meat and the decrease on the health system costs.
- In the material and methods section, I suggest detailing the descriptions of the taxes applied to reduce GHG emissions. For example, are taxes applied to the supply or demand of products?
- In section 2.2, I suggest that the mathematical model be presented in a summarized way in the methodology and in detail in the annexes. To better understand the results and conclusions it is necessary to show the main mathematical equations.
- In the results and conclusions, the authors can also discuss about the possibilities of using the financial resources collected from the carbon taxes. These resources can be redirected to encourage new clean technologies, improvements in the health system, among other aspects.
Reviewer 2 Report
Review: Can the greenhouse gas emissions tax on food also be healthy and equitable? A systematic review and modelling study from Aotearoa, New Zealand
The subject of the study is current and relevant. First, however, the authors must clarify a set of concepts, analytical scopes and methodological aspects.
Objective and scope: The objective of the presented study is "was to estimate the impact of greenhouse gas (GHG) food taxes and assess whether such a tax could also have health benefits in Aotearoa NZ.". However, no model was applied that allows estimating impacts/effects to confuse the reader. The study does a systematic literature review within the PRISMA approach. It is essential to make this clear.
The relationship between systematic literature review and tax scenario construction is confusing. Therefore, the authors need to explain the introduction and methodology section more clearly.
The review of modelling studies on food tax policies needs to be further discussed regarding the link with New Zealand. The linkage between the review and the NZ context is poor, confusing, and non-specific.
The authors mention in the abstract and the introduction that they estimate the impact of taxes on several outcomes (population health gain (in quality-adjusted life-years: QALYs), health system cost-savings, health inequities between Māori (NZ indigenous population) and non -Māori, GHG emissions and food costs). However, no research problem/question elements support the need to generate evidence for such a broad set of outcomes. In the end, one has the feeling that the authors do not have enough arguments to qualify the research problem. How these different outcomes are linked to the study's objective is not clear. How these outcomes are associated with the reality of NZ is not clear either. It is crucial to qualify the contextualization and justifications for the approach and scope of the study.
In the conclusions, the authors mention: "Applying consumption taxes that target foods with high GHG emissions associated with their production has the potential to result in population health and climate co-benefits alongside potential savings to the health system and reductions in ethnic health inequities. Whilst NZ is used as a case study, findings are likely to be of relevance to other high-income countries. Policymakers need to weigh up the benefits of such a consumer food tax against other options that are likely to result in improved diets and reduced food system GHG emissions. The consumption tax combined with a subsidy on fruit and vegetables could be considered to minimize the impact on household food costs."
However, the link between the results obtained and these conclusions is unclear. The vagueness of the conclusions evidences a lack of in-depth study of the study's results. Adding to the lack of a research problem, this could invalidate the analysis of such a wide set of outcomes, since in the conclusions, this simply disappears.
Reviewer 3 Report
The publication of the paper is justified because the topic is treated in an original and appropriate way thanks to the development of issues related to healthy, food, and greenhouse tax emission through an interesting research. Literature is developed in a clear way, but could be further expanded citing the main works about the themes treated.
The methodology of the empirical research is developed and explained in a understandable way also for a wide public responding appropriately with the paper objectives. Good presentation of the results, because the paper presents an important coherence between the objectives of introduction and methodology and the discussion.
Also the conclusions are understandable but could be more expanded in response to the objectives set by the paper.
